# Recent Advances in Encapsulation Techniques of Plant Growth-Promoting Microorganisms and Their Prospects in the Sustainable Agriculture

**Amel Balla** [1], **Allaoua Silini** [1], **Hafsa Cherif-Silini** [1], **Ali Chenari Bouket** [2], **Faizah N. Alenezi** [3] **and Lassaad Belbahri** [4,*]

1 Laboratory of Applied Microbiology, Department of Microbiology, Faculty of Natural and Life Sciences, University Ferhat Abbas of Setif, Setif 19000, Algeria
2 East Azarbaijan Agricultural and Natural Resources Research and Education Centre, Plant Protection Research Department, Agricultural Research, Education and Extension Organization (AREEO), Tabriz 5355179854, Iran
3 Marine Biodiscovery Centre, Department of Chemistry, University of Aberdeen, Old Aberdeen, Scotland AB24 3UE, UK
4 Laboratory of Soil Biology, University of Neuchatel, 11 Rue Emile Argand, CH-2000 Neuchatel, Switzerland
* Correspondence: lassaad.belbahri@unige.ch

**Abstract:** In addition to changing global demography and global warming, agricultural production systems around the world are threatened by intensive agricultural practices (overuse of land and excessive use of chemical fertilizers and pesticides) that deplete soils by affecting their dynamics and their fertility, pollute the environment, lower production, and alter biodiversity on a large scale. The use of bioformulations based on PGPMs (plant growth-promoting microorganisms) seems to be a promising and sustainable strategy to overcome these threats, thanks to their tolerance to various biotic and abiotic stresses and via their beneficial effects in promising plant growth, pest protection, bioremediation, and restoration of degraded lands. In recent years, particular attention has been paid to encapsulated formulations because they offer several advantages over conventional bioformulation (liquid and solid) related to shelf life, problems of survival and viability in the environment, and the efficiency of rhizospheric colonization. This review focuses on the types of encapsulations and the different technologies used in this process as well as the most commonly used substrates and additives. It also provides an overview on the application of encapsulated bioformulations as biofertilizers, biopesticides, or other biostimulators and summarizes the knowledge of the scientific literature on the development of nanoencapsulation in this sector.

**Keywords:** PGPMs; bioformulation; encapsulation; sodium alginate; nanotechnology

## 1. Introduction

Current agricultural activities, through the excessive and uncontrolled use of toxic and harsh fertilizers and pesticides, cause significant economic and ecological damage and risks to animal and human health [1]. In recent decades, the use of PGPMs (plant growth-promoting microorganisms) in the form of inoculants as alternatives to chemicals has received increasing attention from researchers in the field due to their ability to stimulate plant growth and protect against pathogen attacks and stressful abiotic factors [2]. Adverse environmental conditions and the presence of toxic compounds and competition with native flora limit the effectiveness and performance of PGPMs [3]. In order to develop a protective tool, various formulations have been designed depending on the application conditions. Immobilization of PGPMs inside biodegradable polymers or encapsulation is a promising bioformulation that preserves their activities related to plant growth promotion, thus leading to maximum cell viability and survival and increased colonization of the rhizosphere and roots of plants [3]. The objective of this review is to highlight the different

encapsulation techniques of PGPMs and the most used carriers and additives and evaluate the work on the application of encapsulated PGPMs in the agricultural sector.

## 2. Limitations of Conventional Bioformulation and the Development of Bioencapsulation

Bioformulation is biologically defined as all active products containing one or more beneficial microbes or their metabolites and which are immobilized on an inert carrier material [4]. The microbe is a living organism and considered a bioagent and can be either a single strain, a spore, or a microbial consortium [5]. Several microorganisms are used to develop bioformulations, namely, bacteria, actinobacteria, and fungi [5]. The carrier material is an inactive material and serves as a carrier of the living organism, providing a protective niche. This carrier must be nontoxic, be chemically stable, be readily available, be inexpensive, be able to maintain humidity, ensure the viability of cells during storage, and ensure their transport close to the target plant [6]. The medium used can be organic (soil, peat, coal, etc.) or inorganic (polymers), solid or liquid [3]. Alongside the support and the living organism, bioformulations may also contain additives, such as starch and humic acid, which provide nutritional support to the formulated microorganisms [7]. Whether solid or liquid, conventional bioformulation has been for a long time the technology of choice for the development of bioinoculants [8]. On a practical level, the techniques are easy to develop, and the materials used are most often available [3].

Solid bioformulations are made by mixing the beneficial microorganism with a solid carrier, which is used as a vehicle. Peat is the preferred carrier in this type of formulation. With its large surface area, high water retention capacity, and composition, it provides a favorable environment for metabolic activity and cell multiplication during storage [5]. However, peat compounds can affect the development of certain microorganisms [9]. Besides peat, various other media are used, such as biochar, bagasse, cork compost, attapulgite, sepiolite, perlite, and amorphous silica [3,10]. Solid bioformulations include granules, microgranules, wettable powders (WPs), wettable granules (WGs)/water dispersible granules (WDGs), and dusts [5]. Liquid bioformulations are usually microbial suspensions in water, in oil, or sometimes in both [11]. This type of formulation allows the formulated microorganisms to quickly come into contact with the target plant and exert their beneficial actions there. However, the absence of the support makes it sensitive to contamination and to prolonged storage conditions and periods [3]. Besides the carrier liquid and the microorganism, dispersing agents and surfactants can be added to the formulations in order to improve their physicochemical quality [12]. Suspension concentrates, oil miscible flowable concentrates, ultra-low-volume suspensions, and oil dispersions are the most reported types of liquid bioformulations [5]. Liquid bioformulations are applied directly to seeds, while solid bioformulations are even applied as a soil amendment [13].

Due to the high demand for bioinoculants based on PGPMs in the agricultural sector and in order to overcome the limitations linked to conventional bioformulations (solid and liquid), science has moved towards the development of new formulations, ensuring better viability, more high stability during storage and transport, ease of use, and better performance in the field [14]. Not only that, these formulations must guarantee the success of the application under extreme conditions, such as saline, acidic, or alkaline soils; high temperatures; and drought [14]. Immobilized formulations of PGPMs in polymers or encapsulation are an advanced, promising, and rapidly developing technology that has significant advantages over other formulations. Encapsulation tends to cover living cells with a polymer matrix that serves as a microenvironment that protects cells and their metabolites from adverse external conditions. This process is also called immobilization. Comobilization is when multiple strains are applied [3]. In the agricultural field, the advantages of bioencapsulation over other bioformulations are crucial. In addition to the large number of cells trapped in formed capsules, this process stabilizes the viability of microbial cells during and after encapsulation, especially during long periods of storage or even when applied to crops; maintains the effectiveness of the properties of activities related to the promotion of long-term plant growth; and ensures the uniform and gradual

controlled slow release of cells near the targeted site, for effective rhizospheric and root colonization [15,16]. Bioencapsulation is generally carried out in three distinct steps:

In the first step, the active principle or the microorganism is mixed and absorbed in a polymeric matrix. Then in the second step, which is a mechanical operation, a liquid solution is dispersed under agitation, where solid particles are formed. In the third step, the particles formed during the previous step undergo polymerization and physicochemical stabilization [3]. Depending on the size of the particles generated, which varies from a few nanometers to a few millimeters, this method is divided into macro-, micro-, and nanoencapsulation.

Encapsulation in macroparticles (1–4 mm in diameter) is a promising technology, especially in developing countries, because it does not require special equipment or materials, and the required processes are generally available [17]. However, this technology faces two main drawbacks: (1) When the capsules are randomly mixed with the seedlings, they can position themselves a few centimeters away from the seed, which causes the microorganisms released from the capsule to migrate to the seed while resisting adverse soil conditions and native microflora and predators. (2) Additional inoculation during planting is recommended despite the bolls being premixed with the seedlings (Figure 1) [17].

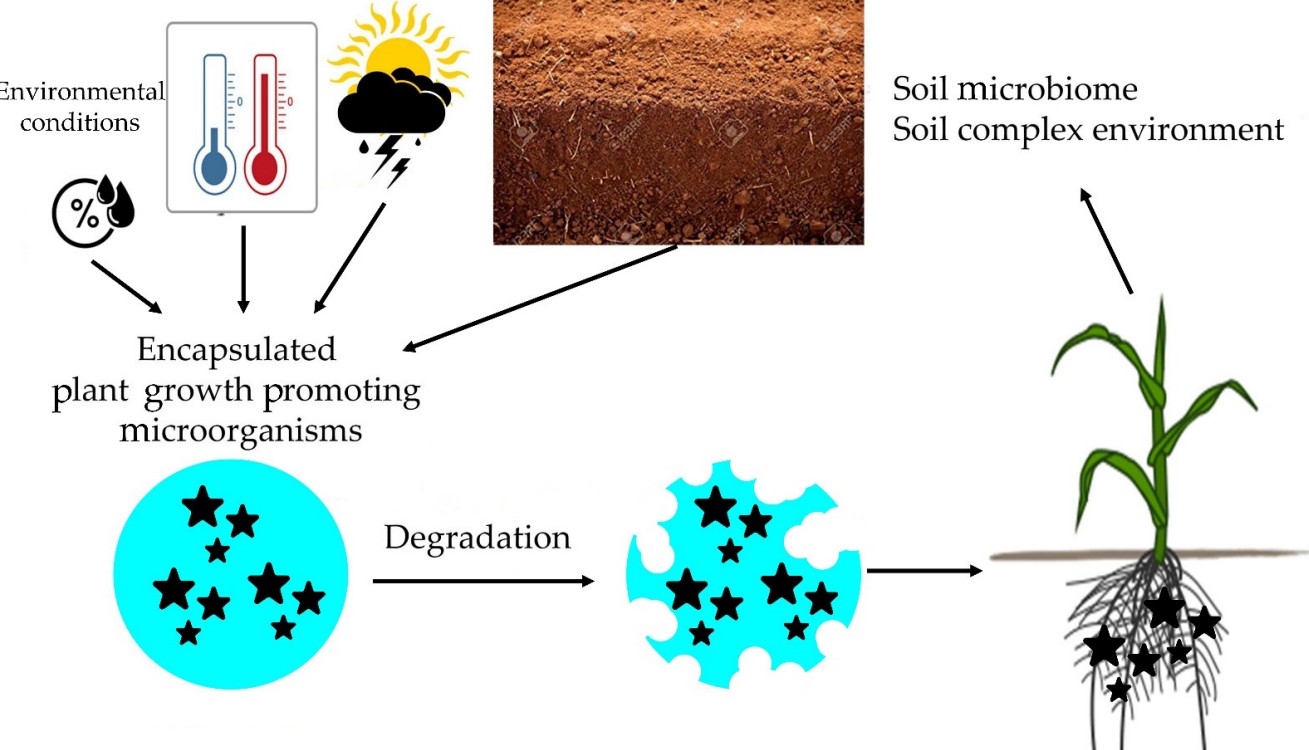

**Figure 1.** Encapsulation of PGPMs and how to use them in plant fertilization.

Formulations immobilized in microparticles (1 mm in diameter) seem to be able to overcome the disadvantages of macroencapsulation and ensure a higher survival rate and a better performance in the field [17]. Although the technology of microbeads is not new [18], it still has two major shortcomings, which are the relatively high cost of the materials required and the specific apparatus necessary for the development of the process [6].

## 3. Encapsulation Techniques

Several encapsulation techniques by physicochemical processes are described in the literature, and others continue to emerge in order to develop new formulations. According to Hudson and Margaritis [19], there are 20 techniques for the elaboration of polymeric capsules, each of which has these uses, requires different materials and equipment, and

has advantages and disadvantages: (1) external gelation, (2) emulsification and internal gelation, (3) emulsion cross-linking method, (4) reverse microemulsion technique, (5) emulsion–solvent extraction, (6) emulsification solvent diffusion method, (7) emulsion-droplet coalescence method, (8) complex coacervation, (9) reverse micellar method, (10) self-assembly methods, (11) water-in-oil emulsification, (12) desolvation process, (13) pH coacervation method, (14) emulsification, (15) nanoparticle albumin-bound (nab) technology, (16) self-assembly, (17) desolvation method, (18) methods involving hydrophobized pullulan derivatives, (19) reverse micelle synthesis method, and (20) emulsification–diafiltration. Ionic gelation (extrusion or cross-linking), emulsification, and spray drying techniques are most often used for the encapsulation of PGPM (Figure 2) [20,21].

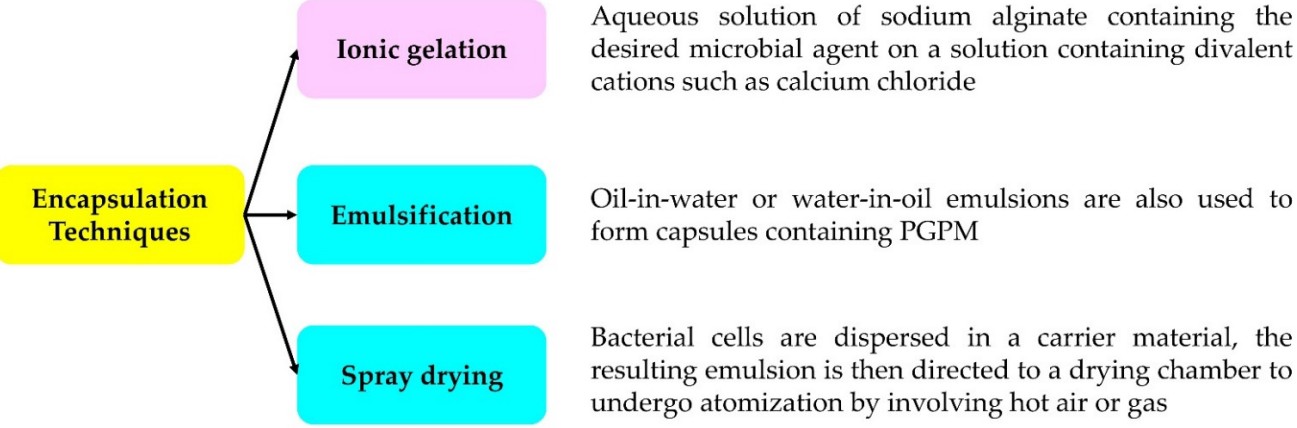

**Figure 2.** Diagram of encapsulation techniques of plant growth-promoting microorganisms.

### 3.1. Ionic Gelation

Ionic gelation is a highly appreciated technique used worldwide due to affordable production costs and accessible methodological conditions [22]. This technique consists in dispersing an aqueous solution of sodium alginate containing the desired microbial agent on a solution containing divalent cations, such as calcium chloride. A hydrogel is then formed after solidification of the droplets after interaction between the polymer chain of the charge negative content of sodium alginate and $Ca^{2+}$ cations [21]. This technique generates uniform beads whose size can vary from a few micrometers to several millimeters, depending on the needle size, and can therefore be adapted to macroencapsulation and microencapsulation. In addition, nanoscale beads are formed by special encapsulation devices [23]. Other polymers, such as pectinate derivatives and guar gum, are used as an alternative to sodium alginate, while calcium gluconate, $Ba^{2+}$, and $Cu^{2+}$ are used to replace $CaCl_2$ [20].

### 3.2. Emulsification

Oil-in-water or water-in-oil emulsions are also used to form capsules containing PGPMs. Polymers are suspended in water and mixed with immiscible oils. The microbial suspension is then added with stirring [21]. This technique gives microcapsules whose size differs according to the emulsification method and the shaking speed, while the size distribution is generally higher than that of the drip techniques [21]. Various emulsification methods are described in the literature, such as thermal gelation and interfacial polymerization [21]. Polymers such as sodium alginate, gelatin, or gum [24] and oils such as soybean oil and paraffin oil are the most used materials in emulsification techniques [25,26].

### 3.3. Spray Drying

Spray drying is a well-known and common dehydration process for the production of marketable immobilized microbial inoculum [27]. During this process, bacterial cells are dispersed in a carrier material. The resulting emulsion is then directed to a drying chamber

to undergo atomization by involving hot air or gas. The ventilation of the extractor generates by the following evaporation of the solvent to leave only powder, which is actually dry microcapsules [24]. Although spray drying is a cost-effective method and provides stable, quality, and rapidly soluble microcapsules, it exhibits a number of disadvantages, particularly in the choice and selection of the material, which must be of low viscosity and high concentration and solubility and have infallible drying characteristics [28]. In addition to the material, the high drying temperature is a key parameter that influences the encapsulation and survival of bacterial cells during the process. This requires a controlled optimization of the inlet and outlet temperature and an adequate material in order to have stable, viable, and homogeneous capsules [27]. Chi et al. [29] found that the number of *Bacillus megaterium* bacteria in microcapsules obtained by spray drying was $10^{10}$ CFU/mL, which was maintained after 6 months of storage at 4 °C. Another team succeeded in obtaining $10^8$ CFU of the biocontrol agent *Streptomyces fulvissimus* per gram of microcapsules by the same technique [30], while Costa et al. [31] claimed a high survival and viability of a strain of *Pantoea agglomerans* after rehydration of the capsules obtained by spray drying.

## 4. Choice of Polymeric Carriers

It has already been established that the quality and effectiveness of the formulation is largely influenced by the carrier. The latter is the predominant constituent of the formulation, which will transport the living microorganism and in the desired concentration to the field while preserving its performance [13]. According to Bashan et al. [13], an ideal carrier should have the following characteristics: (i) is biodegradable and nontoxic and poses no risk to humans, animals, and the environment; (ii) allows the preservation of the microorganism and its performance over a long period of storage; (iii) is available at a reasonable cost and easy to handle and ensures controlled release of the microorganism; (iv) is easy to manufacture and to combine with additives or nutrients; and (v) is adapted to PGPM strains and has physicochemical characteristics allowing a high-water retention capacity.

### 4.1. Sodium Alginate

Sodium alginate is a polysaccharide abundant in the cell walls of brown algae, and also present in the walls of certain bacterial species, such as *Pseudomonas* and *Azotobacter* [32]. This polymer consists of β-d-mannuronate and α-l-guluronate residues whose carboxylate groups carry a net negative charge [32]. Alginate is applied in many fields, namely, pharmaceutical, food, and agricultural [33,34]. Among its applications, it is considered the substrate of choice for the encapsulation of PGPMs in consideration of its biodegradability, biocompatibility, availability, nontoxic nature, relatively low cost, and ability to withstand acidic soil conditions and trap a large number of microbial cells and allow their slow and progressive diffusion [35,36]. Water-soluble alginate forms an irreversible hydrogel after reacting with acids or salts containing divalent cations [37]. Guluronic-acid-based alginate has more ion affinity than mannuronic-acid-based alginate, and the gel matrix is formed after diffusion of cations into the alginate solution after exchange of $Na^+$ with $Ca^{2+}$ and the formation of an ion bridge between the two chains [38]. However, it has been reported that the presence of certain antigelling agents, such as $Mg^{2+}$, or chelating agents, such as citrate, can affect the integrity of alginate capsules [39]. The combination of alginate with other polymers, such as chitosan and gelatin, seems to be the solution [24].

### 4.2. Chitosan

Chitosan can also be used as a formulation carrier for plant-beneficial microorganisms [40]. Chitosan is made up of long chains of N-acetyl glucosamine (GlcNAc) units. It also includes N-glucosamine (GlcN) units, which are more abundant than GlcNAc [41]. This oligosaccharide shares the same characteristics of biodegradability, nontoxicity, ease of handling, and low cost with alginate [42]. Chitosan can also induce the production of osmoregulators in plants and exhibit significant antimicrobial activities [7]. The major drawback of using chitosan as a carrier for the encapsulation of microbes is its limited

mechanical resistance and its low chain flexibility [43]. Its combination with other polymers, such as alginate and starch, can improve these characteristics by strengthening its structure and physicochemical stability [44]. In a recent study, chitosan at different concentrations (0.3–3%) was used in combination with 2% alginate for the encapsulation of *Methylobacterium oryzae*, with a formulation-maintained viability of $10^7$ CFU/mL and 80% survival after 3 months of storage. The application showed better promotion of tomato seedling growth compared with formulations of alginate alone [40]. Another research team successfully encapsulated *Azospirillum brasilense* and *Pseudomonas fluorescens* in a formulation of chitosan (3%) and starch (8%). The formulation ensured the viability of $10^9$ CFU/g for *A. brasilense* and $10^8$ CFU/g for *P. fluorescens* after 12 months of storage at room temperature [45].

*4.3. Chitin*

Chitin is another biopolymer used in encapsulation as a filler [7]. Chitin is the second most abundant biomaterial in nature after cellulose, present especially in yeasts, fungi, insects, and marine invertebrates [46]. It is a homopolymer of 2-acetamido-2-deoxy-β-D-glucopyranose. Chitin is called chitosan when the free amine form is more than 50% deacetylated [46]. Chitin also exhibits significant antimicrobial properties and is used as a seed coating agent; this appears to stimulate the secretion of plant chitinases to control the invasion of harmful pests [46]. The addition of chitin in an alginate-based bioformulation induced chitinase synthesis in *Penicillium janthinellum* and improved its ability to solubilize phosphate in [7]. Chitin is insoluble in most common solvents, which limits its use in certain encapsulation processes [20] (Figure 3).

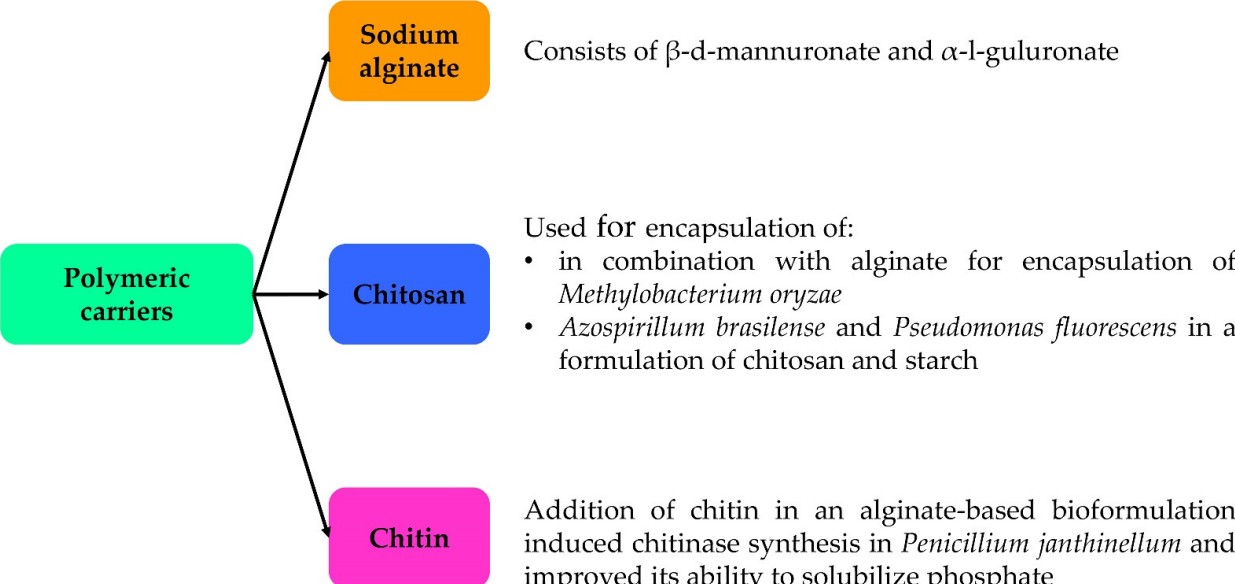

**Figure 3.** Polymeric carriers used in the processes of PGPM encapsulation.

## 5. Choice of Additives

In addition to the carrier polymers necessary in the encapsulation processes, additives are added to the formulations in order to reduce their cost and to improve certain characteristics, such as survival and performance in the field [7], and to maintain the stability of the cells inside the beads during formulation, storage, and transportation to application fields [6], which helps to control release and quickly adapt to the application environment. There are a wide range of additives, such as starch, clay, humic acid, skimmed milk, and sugars [7]. However, the choice of the most adequate additive that suits the types of encapsulated cells and the carrier used is critical and needs to be well studied (Figure 4) [6].

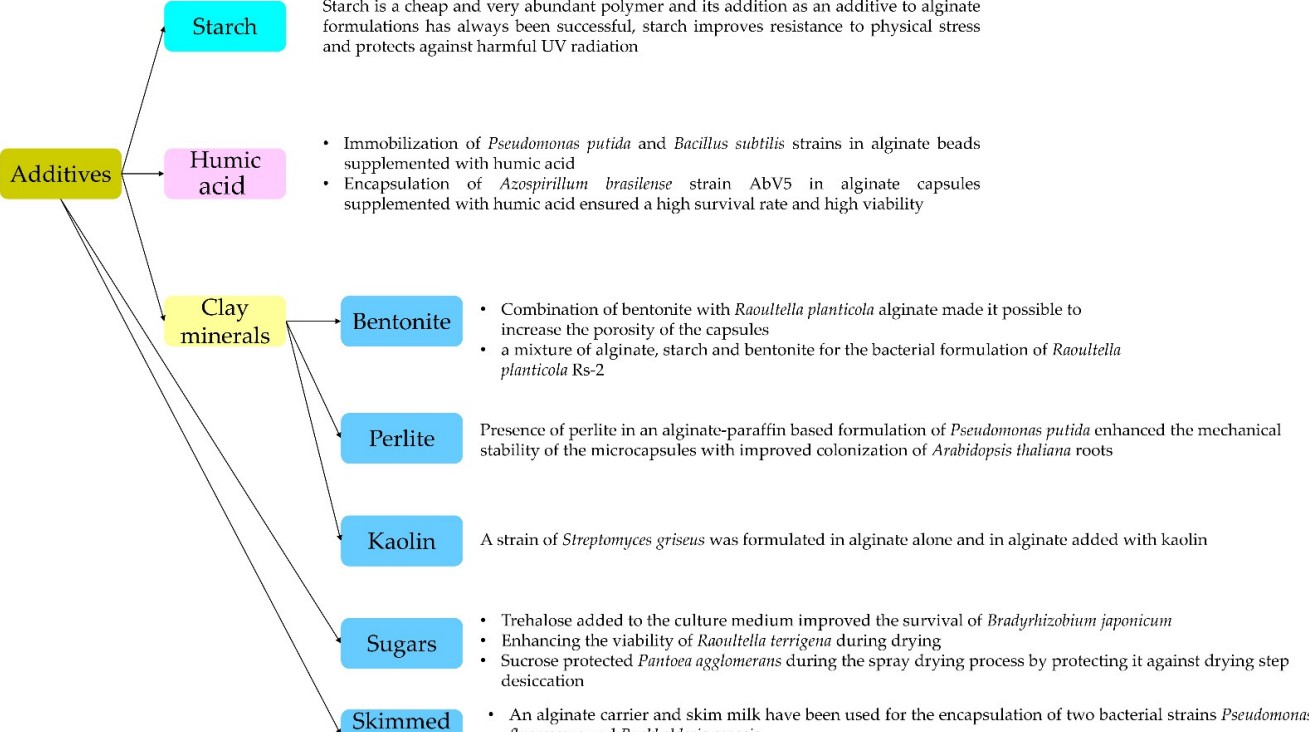

**Figure 4.** Additives used in the process of encapsulation of PGPMs.

### 5.1. Starch

Starch contains two types of alpha-glucans, which are amylose and amylopectin whose concentrations vary according to the botanical origin of the starch [47]. Starch is a cheap and very abundant polymer, and its addition as an additive to alginate formulations has always been successful. Starch improves resistance to physical stress and protects against harmful UV radiation [18,48]. In addition, it has a protective effect, thanks to cell adhesion to its granules. This protects bacteria from stress particularly during the critical drying stage, and helps maintain a stable microbial population over a long storage period [49]. In addition, high levels of starch increase the viscosity and decrease the porosity of the capsules, which allows a more controlled diffusion of microorganisms in the environment [7]. The alginate–starch combination is perfect, especially for microorganisms that can use starch as a carbon source; this finding is confirmed by electron microscopy images that show a homogeneous distribution between the particles of the two polymers [49].

### 5.2. Humic Acid

Humic substances constitute the major element of the organic matter of soil, water, and sediments [50] whose humic acid assembles various heterogeneous components of low molecular weight [51]. The exact chemical composition of humic acids is difficult to establish and differs according to several geographical climatic and biological parameters. Humic acids generally include phenolic, carboxylic acid, enolic, quinone, and ether functional groups, sugars, and peptides [52] The presence of humic acid in the bulk soil of the plant or in the rhizosphere has been shown to be very favorable; it accelerates the absorption of nutrients, carbon, and nitrogen in addition to the induction of secondary metabolism [53]. Humic acid also promotes the formation of lateral roots and root hairs, as it stimulates chemotaxis within the microbial community of the rhizosphere, thus leading to efficient rhizospheric and endophytic colonization [50]. Humic acid has already demonstrated a significant beneficial effect on different crops, including sugarcane, tomato, common beans, and maize [50].



Humic acid forms strong chemical bonds with alginate, resulting in stable, high-performance capsules. Immobilization of *Pseudomonas putida* and *Bacillus subtilis* strains in alginate beads supplemented with humic acid applied to *Lactuca sativa* seedlings showed a significant increase in shoot and root height after inoculation compared with the free control [54]. Thus, the addition of 1% of humic acid in 2% of alginate formulation of a strain of *Bacillus subtilis* made it possible to maintain high viability after 5 months of storage and to ensure successful promotion of the growth of lettuce plants under gnotobiotic conditions [52]. Meanwhile, the encapsulation of the *Azospirillum brasilense* strain AbV5 in alginate capsules supplemented with humic acid ensured a high survival rate and high viability after 90 days of storage [55].

### 5.3. Clay Minerals

Clay minerals are the most ubiquitous minerals on the biosphere. These minerals have exceptional physicochemical characteristics, in addition to their high water retention capacity and their high carbon and potassium content. The biogeochemical activity of microorganisms results from their interaction with these minerals; the latter are involved in the growth and metabolic activity in microbial and various other microbial processes [56]. Bentonite, perlite, and kaolin are the clay minerals most commonly used as additives in PGPM encapsulation processes.

#### 5.3.1. Bentonite

Bentonite is widely used in the pharmaceutical industry and in agriculture as a vector for drugs and pesticides [57]. Zohar-Perez et al. [58] deduced that the use of 0.5% of bentonite and 0.5% of kaolin in combination with 2% of alginate increases the thickness of the capsule walls, which leads to maintaining a high survival rate after exposing them to UV radiation. The combination of bentonite with *Raoultella planticola* alginate made it possible to increase the porosity of the capsules and, therefore, to regulate the diffusion of the microorganism in the soil in [59]. Wu et al. [60] used a mixture of alginate, starch, and bentonite for the bacterial formulation of *Raoultella planticola* Rs-2. This combination was found to be better for controlled release under abiotic stress conditions. The combination of bentonite with starch was designed for the immobilization of *Pseudomonas putida* Rs-198 in alginate; this mixture increased the survival rate during storage. Furthermore, the application of this formulation to cotton plants under salt stress led to more enhanced root colonization and increased biomass and soluble protein content [61].

#### 5.3.2. Perlite

Perlite is a porous and structurally amorphous natural inorganic material that has very high physical and biological chemical resistance [62]. Perlite was used as an alternative to peat for the formulation of rhizobia strains; this material made it possible to maintain a high percentage of survival after storage at 25 °C for 120 days and to boost the performance of rhizobia in the promotion of growth of soybean plants [10]. According to Sari et al. [63], the addition of perlite in polymer-based bioformulations provides better mechanical stability and protects against microbial threats and against unfavorable physicochemical conditions. In addition, the presence of perlite in an alginate–paraffin-based formulation of *Pseudomonas putida* enhanced the mechanical stability of the microcapsules with improved colonization of *Arabidopsis thaliana* roots after 21 days of colonization in [26].

#### 5.3.3. Kaolin

The chemical composition of Kaolin rich in kaolinite makes it one of the most important clay minerals in the industry [64]. A strain of *Streptomyces griseus* was formulated in alginate alone and in alginate (2.5%) added with kaolin (1:4 kaolin/alginate). The addition of this mineral not only improved the effectiveness of the formulation but made it possible to effectively control a phytopathogenic strain of *Fusarium oxysporum* f.sp. *cubense* tropical race 4 compared with the alginate formulation alone [65]. In addition, a freeze-dried

formulation of *Streptomyces* sp. based on alginate and kaolin effectively suppressed the pathogenic fungus *Rhizoctonia* on tomato plants in [66].

### 5.4. Sugars

Several sugars are used in bioencapsulation technologies as external protectants [7,67]. Sugars, such as trehalose, sucrose, glucose, and fructose, protect against osmotic pressure and desiccation during the drying step [68]. Sugars can be added to formulations after harvesting cells or added to culture media as they have the potential to be absorbed and aggregated in the cytoplasm [67]. The protective effect of trehalose consists in maintaining an intact and fluid membrane, thanks to the hydrogen bonds formed with proteins under desiccation conditions [69]. Trehalose added to the culture medium improved the survival of *Bradyrhizobium japonicum* [70] and enhanced the viability of *Raoultella terrigena* during drying [47]. Sucrose protected *Pantoea agglomerans* during the spray-drying process by protecting it against drying step desiccation [71], while glucose in combination with skimmed milk and polyvinylpyrrolidone K-90 provided 78% survival of *Beauveria bassiana* after 12 months of storage at 30 °C [72]. Other mono- and disaccharides are also used as protectors, such as fructose and lactose [71].

### 5.5. Skimmed Milk

Skimmed milk is also used as a protective agent and as a source of nutrients in bioformulations [7]. Several studies have reported the use of skimmed milk in the immobilization of PGPM. An alginate carrier and skimmed milk have been used for the encapsulation of two bacterial strains, *Pseudomonas fluorescens* and *Burkholderia cepacia*, and the application of their microcapsules improved wheat plant growth under semiarid conditions in [73]. In another study, *Pseudomonas fluorescens* was immobilized in alginate and skimmed milk, which resulted in an effective release of the microorganism into the soil [74]. Thus, skimmed milk in combination with clay boosted the performance of *Enterobacter* sp. under soil conditions [7]. Furthermore, Bashan et al. [18] found that skimmed milk had a significantly positive effect on the number of *Azospirillum brasilense* cells trapped and on the efficient release into the adjacent soil of the plant. Many other materials are used as additives in microorganism encapsulation technology, such as protein hydrolysates, glycerol, silicon, polyactic acid, strigolactones, carboxymethylcellulose (CMC), sorbitol, polyethylene glycol (PEG), sodium glutamate, mannitol, and gelatin [7,67]. The addition of these additives largely depends on the encapsulation method used, the type of carrier used, and the microorganisms formulated.

### 6. Recent Advances in Encapsulated Biocontrol Agents

Over the past two decades, many research results have been obtained on the application of encapsulated PGPMs for biological control purposes (Table 1). Articles published in leading journals have reported the success of inoculation of plants by beneficial microorganisms. Guo et al. [75] showed that the encapsulated coculture of *Klebsiella oxytoca* and three strains of *Bacillus subtilis* is very promising in the biological control of *Rhizoctonia solani* and in the mitigation of the effect of salinity. The application of the charged coculture microcapsules on cotton plants improved the physio-biochemical parameters of the plant and reduced the rate of fungal infection and antioxidant enzymes. Pour et al. [25] investigated the encapsulation of *Pseudomonas fluorescens* in alginate–chitin microbeads supplemented with $SiO_2$ nanoparticles. This formulation, when applied on *Fusarium solani*–infected potato seeds, reduced infection by 75% and significantly increased morphological parameters (the fresh and dry weight, the height of roots and shoots, and root volume). Thus, in greenhouse experiments on wheat seedlings, microcapsules of *Streptomyces fulvissimus* gave 90% control of take-all disease caused by *Gaeumannomyces graminis* [30]. The *Pantoea agglomerans* strain E325 was encapsulated in 0.8–2% of alginate microcapsules, and its efficacy in the biocontrol of *Erwinia amylovora* was evaluated. The results indicated that the encapsulated cells significantly reduce the incidence of *Erwinia amylovora* in apple infection and may be a good

alternative to chemical pesticides in the management of plant pathogens [16]. The coating of tomato seeds with microcapsules loaded with the plant growth-promoting actinobacteria *Streptomyces* sp. Di-944 successfully controlled *Rhizoctonia solani*, the causative agent of damping off. The coating effect was similar to oxine benzoate fungicide and significantly superior to the commercial biological control agent *Streptomyces griseoviridis* [66].

**Table 1.** Examples of characteristics of polymeric formulations of PGPM and their application purposes.

| Microorganism | Carrier | Additives | Method | Purpose | Plant | References |
|---|---|---|---|---|---|---|
| *Kosakonia radicincitans* | Amidated pectin | Maltodextrin Sorbitol monosodium glutamate | Cross-linking (extrusion) | Osmoprotection and desalination | Radish | [76] |
| *Raoultella planticola* | Sodium alginate | Bentonite | Cross-linking extrusion) | Biofertilizer | - | [59] |
| *Pseudomonas putida* | Sodium alginate + paraffin | Bentonite | Emulsification (external gelation) | Plan growth promotion | - | [77] |
| *Pantoea agglomerans* | Sodium alginate | - | Cross-linking (extrusion) | Desalination | Rice | [78] |
| *Methylobacterium oryzae* | Sodium alginate + chitosan | - | Cross-linking (extrusion) | Seed germination and plant growth promotion | Tomato | [40] |
| *Klebsiella oxytoca* + *Bacillus subtilis* | Sodium alginate | - | Cross-linking (extrusion) | Biocontrol of *Rhizoctonia solani* under saline conditions | Cotton | [75] |
| *Pseudomonas fluorescens* | Sodium alginate | - | Cross-linking (extrusion) | Polychlorinated biphenyl degradation bioremediation | - | [74] |
| *Pseudomonas fluorescens* | Sodium alginate + soybean oil | Gelatin | Emulsification (internal gelation) | Biocontrol of *Fusarium solani* | Potato | [25] |
| *Trichoderma viride* | Sodium alginate | - | Cross-linking (extrusion) | Plant nutrition | - | [79] |
| *Beauveria bassiana* | Sodium alginate | Bentonite | Cross-linking (extrusion) | Biocontrol | - | [80] |
| *Streptomyces fulvissimus* | chitosan + gellan gum | - | Spray drying | Biocontrol of *Gaeumannomyces graminis* | Wheat | [24] |
| *Bacillus subtilis* | Sodium alginate | Humic acid + glycerol | Cross-linking (extrusion) | Plant growth promotion | Lettuce | [52] |
| *Bacillus megaterium* | Chitosan + maltodextrin | Glucose, sucrose, skimmed milk powder, trehalose, lactose, arabic gum, gelatin, modified starch, sodium alga acid, and β-cyclodextrin | Spray drying | Bioremediation of salinized soils | - | [29] |
| *Azospirillum brasilense* | Sodium alginate | Humic acid + trehalose + peat | Cross-linking (extrusion) | Plant growth promotion | Wheat | [55] |
| *Pantoea agglomerans* | Sodium alginate | - | Cross-linking (extrusion) | Biocontrol of *Erwinia amylovora* | Apple | [16] |
| *Pseudomonas putida* | Sodium alginate + paraffin | Perlite | Emulsification | Plant growth promotion | *Arabidopsis thaliana* | [26] |
| *Pseudomonas fluorescens* + *Pseudomonas putida* | Eudragit + methacrylic copolymer | Silica | Spray drying | Biofertilizer | - | [81] |
| *Streptomycetes* sp. | Sodium alginate | Kaolin + starch + talc | Cross-linking (extrusion) | Biocontrol of *Rhizoctonia solani* | Tomato | [66] |
| *Beauveria bassiana* | Sodium alginate | Peanut oil | Cross-linking (extrusion) | Biocontrol of *Solenopsis invicta* | None | [82] |

**Table 1.** *Cont.*

| Microorganism | Carrier | Additives | Method | Purpose | Plant | References |
|---|---|---|---|---|---|---|
| *Bacillus subtilis* + *Pseudomonas corrugata* | Sodium alginate | Skimmed milk | Cross-linking (extrusion) | Plant growth promotion | Maize | [83] |
| *Pseudomonas fluorescens* + *Burkholderia cepacia* | Sodium alginate | Skimmed milk | Cross-linking (extrusion) | Biofertilizer in salinized soil | Wheat | [73] |
| *Klebsiella oxytoca* | Sodium alginate | - | Cross-linking (extrusion) | Biofertilizer in salinized soil | Cotton | [84] |
| *Sinorhizobium meliloti* | Canola oil + xanthan gum | - | Emulsification | Nodulation and plant growth promotion | Alfalfa | [85] |
| *Bacillus subtilis* | Alginate | Bentonite + starch + titanium dioxide nanoparticles | Cross-linking (extrusion) | Biocontrol of *Rhizoctonia solani* and plant growth promotion | Beans | [86] |
| *Pseudomonas putida* + *Bacillus subtilis* | Sodium alginate | Humic acid | Cross-linking (extrusion) | Plant growth promotion | Lettuce | [54] |

## 7. Recent Advances in Encapsulated Biofertilizers and Growth Stimulator Agents

It has been well established that the introduction of free or bioformulated PGPMs into the soil increases soil fertility and promotes plant growth and nutrition while preserving the functionality and complexity of native microflora. The bioformulation of fertilizers and growth promoters has been widely studied, and the success of their field application has been reported. He et al. [59] reported that *Raoultella planticola* Rs-2 encapsulated in alginate–bentonite microcapsules could be a low-cost option as a fertilizer, thanks to their slow-release properties and relatively high survival rate. Chanratana et al. [40] found that *Methylobacterium oryzae* CBMB20 formulated in wet chitosan produced a significant effect on the shoot and root length and dry weight of tomato plants, and bacteria encapsulated in microbeads had a better rate of survival after 21 days of application in greenhouse soils. *Methylobacterium oryzae* encapsulated in chitosan microbeads could be a novel and feasible technique for soil fertilization application. Another study demonstrated that the in vivo application to lettuce plants of beads containing the plant growth-promoting bacterium *Bacillus subtilis* CC-pg104 achieved significant growth promotion by increasing shoot length and roots and ensuring effective root and rhizospheric colonization; thus, the humic acid added to this formulation boosted the viability of these cells during storage, ensured progressive cell release, and protected the bacteria against unfavorable environmental factors [52]. In another study, John et al. [85] used canola oil and xanthan gum for the formulation of *Sinorhizobium meliloti*. A significant effect on nodulation was observed when applying an emulsion containing $10^5$ CFU/mL on alfalfa seeds. In addition to improving the nodulation index, the nitrogen fixing rhizobia *Sinorhizobium meliloti* also increased nodule size, plant height, shoot dry weight, and root dry weight.

## 8. Recent Advances in Encapsulated Biosensors and Bioremediation Agents

Some PGPMs, in addition to their ability to enrich plant growth and health and their ability to maintain soil quality and fertility, have great potential for remediation and decontamination of polluted environments [87]. The advantage of encapsulating PGPM in bioremediation is to overcome the limiting effect of contaminants on survival and to protect cells from the adverse conditions of polluted environments [88]. The genetically modified rhizobacterium *Pseudomonas fluorescens* F113 Rifpcb has been used as a biosensor and biorestorer of soils contaminated by polychlorinated biphenyls (PCBs), very dangerous chemical compounds carcinogenic and toxic for the environment. Alginate beads loaded

with *Pseudomonas fluorescens* applied to contaminated soil allowed the detection of PCBs and the restoration of the contaminated site. Immobilization in alginate beads allowed a more controlled release and reduced the risk of unwanted diffusion of the genetically modified strain. This system could facilitate the application of depollution bacteria in contaminated soils [74]. On the other hand, the use of encapsulated microorganisms for the desalination of saline soils has been widely reported. The *Pantoea agglomerans* strain KL isolated from saline soil used as inoculum encapsulated in alginate beads has allowed, when applied to rice plants, for reducing stress caused by salt concentration; increasing length, biomass, and the rate of photosynthetic pigment; and reducing the rate of proline and malondialdehyde. This application helps to decrease sodium accumulation and improves rice growth in the presence of salt [78]. In a similar study, spray-dried microcapsules of *Bacillus megaterium* NCT-2 showed very significant remediation potential when applied in highly saline soil, resulting in a significant reduction of more than 45% in $NO_3^-$, and electrical conductivity was observed [29].

## 9. Nanotechnological Application in PGPMs Bioencapsulation

Nanotechnology or nanoscience refers to the engineering of materials and products at the nanoscale [88]. Nanotechnology has an attractive impact in various disciplines, such as physics, chemistry, medicine, and electronics, but also in biology and agriculture. In the context of sustainable agriculture, the development of nanofertilizers and nanopesticides stimulates their performance and effectiveness, minimizes the risk of environmental pollution, and boosts the reproducibility of the approach [89]. Nanoparticles, such as nanoclays and zeolites, can be added in fertilizer or pesticide bioformulations, serving as a multifunctional tool. Microorganisms encapsulated in nanogel benefit from better nutrient supply, excellent growth rate and survival, increased physiological activity, and timed and precise release, and that meets the biological demands of the target plant or site [89,90]. Nanomaterials can also protect encapsulated microorganisms and plants from pest antagonism and stressful abiotic factors [90]. The choice of adequate nanoparticles is crucial to ensure the success of the nanoformulation and its application. Concentration and toxicity tests are necessary to optimize the formulation [90]. In bioformulations are titania nanoparticles ($TiO_2$), silica nanoparticles, silver nanoparticles (AgNPs), gold nanoparticles, nanozeolites, nano zinc oxide, nano carbon, nano boron, and nano chitosan [90].

Panichikkal et al. [91] demonstrated that the immobilization of *Pseudomonas* sp. DN18 in alginate capsules supplemented with salicylic acid (SA) and zinc oxide nanoparticles (ZnONPs) is a promising and stable biopesticide and biofertilizer delivery tool. SA and ZnONPs nanoparticles showed no bactericidal activity against *Pseudomonas* sp. DN18, and did not alter its IAA (indole acetic acid) production property. This nanoformulation applied to *Oryza sativa* seeds was associated with growth promotion and biocontrol activity against the fungus *Sclerotium rolfsii*. In another study, a coculture of *Pantoea agglomerans* and *Burkholderia caribensis* was immobilized in nanofibers; the latter preserved the viability and plant growth-promoting properties of both encapsulated strains, which showed beneficial effects on the germination of seeds, length, and dry weight of soybean roots [92]. Moreover, the addition of titanium dioxide (TNs) nanoparticles in alginate–bentonite–starch microbeads of the *Bacillus subtilis* Vru1 strain markedly improved its ability to inhibit *Rhizoctonia solani* by up to 90% and that of promoting the parameters of the bean growth (Table 2) [86].

**Table 2.** Examples of characteristics of nanoencapsulations of PGPMs and their application purposes.

| Microorganism | Carrier | Additives | Method | Purpose | Plant | References |
|---|---|---|---|---|---|---|
| *Bacillus subtilis* | Alginate | Bentonite + starch + titanium dioxide nanoparticles | Cross-linking (extrusion) | Biocontrol of *Rhizoctonia solani* and plant growth promotion | Beans | [86] |
| *Pseudomonas* sp. | Alginate | Salicylic acid + zinc oxide nanoparticles | Cross-linking (extrusion) | Biocontrol of *Sclerotium rolfsii* and plant growth promotion | *Oryza sativa* | [91] |
| *Pantoea agglomerans* + *Burkholderia caribensis* | Polyvinyl alcohol | Nanofibers | Electrospinning | Plant growth promotion | Soybean | [92] |

## 10. Current State and Future Perspectives

Inoculation technology and the development of bioinoculants in the field of agriculture is an emerging field of research. Various bioformulations of fertilizers and pesticides have been designed within the framework of the development of sustainable agriculture. The advantage of encapsulation technology over other bioformulations has already been proven as it has been established that microencapsulation is preferable among other types of bioencapsulation. PGPMs' loaded microcapsules can be applied directly to the soil, with seedlings or as a seed coating. Microencapsulation makes it possible to control the number of viable and active cells trapped and the number of cells released to reduce the mortality rate during storage, and the microbeads can be applied directly or after a prolonged period of storage at low or at ambient temperature.

Several encapsulation techniques are adopted for the immobilization of PGPMs or microbes for agricultural application. Extrusion or cross-linking and emulsification are the most compatible with heat-sensitive microorganisms, while spray drying is essentially suitable for heat-resistant and sporulating microorganisms. Thus, for an appropriate control, it is necessary to take into account technical parameters, such as the temperature of the process, its duration, the drying conditions, the availability of materials, and the desired equipment. Several types of microorganisms have been encapsulated, such as Gram-positive bacteria, Gram-negative bacteria, actinobacteria, rhizobia, and fungi. Spore encapsulation is also feasible and promising, especially for fungi and spore-forming bacteria.

The encapsulation of cocultures and combinations of several strains is possible, but the success of this design is challenging. Their coexistence inside the beads in the long term and their impact and that of their metabolites on the indigenous microbial communities of the rhizosphere remain not elaborated. More attention should be paid to the study of microbial interactions inside the capsule and within the plant microbiome.

Natural, eco-friendly and low-cost biopolymers are efficient and profitable as formulation support. Additives prolong survival during storage and serve as a carbon/nutrient source. However, the additive(s) that suits the polymer, the microorganism, and the purpose of the application should be chosen.

The incorporation of nanofibers and nanoparticles in bioformulations has shown relevant results in promoting plant growth and protecting crops. However, for a more appropriate field application, detailed studies on the toxicity and impact of nanoparticles on the environment and on humans are recommended.

Hundreds of studies report the use of immobilized bioformulations in vivo (on seeds or on potted plants in greenhouses or in growth chambers under controlled conditions) but rarely on fields. In light of these facts, future research should focus on:

— Industrial and large-scale production and field application of encapsulated bioformulations;
— Evaluation of the performance of encapsulated PGPMs in extreme environments and the long-term effect of interactions of PGPMs with host plants at the molecular, genetic, and physiological levels;

- In-depth evaluation of the improvement of formulations by combining several additives, where the concentration ratios must be carefully controlled;
- Expanding the target crop type and species variation as the majority of work focuses on cereals, legumes, and some vegetables;
- Determination of the effect of the inoculation of polymeric beads on the plant microbiome, in particular on its functioning and its structure;
- Estimation of the optimization of the cost of the formulations and the possibilities of their marketing.

## 11. Conclusions

The use of bioinoculants in agriculture seems to solve the problems related to the excessive use of chemical fertilizers and pesticides. The encapsulation of bioinoculants based on PGPM is a sustainable method that contributes to agricultural productivity, improving accessibility plant nutrients, increasing tolerance to biotic and abiotic stresses, optimizing soil quality and fertility, and fighting pathogens. Many techniques have been developed in the field, of which ionic gelation is most reported. Sodium alginate remains the polymer of choice for bioencapsulation processes. Efforts have also been devoted to the choice of additives, such as starch and clay minerals, and the incorporation of nanotechnologies in this process. Efforts have been devoted to the development of encapsulation technologies, particularly in the choice of polymers and additives and in the incorporation of nanotechnology in this process. Further efforts should be devoted to popularizing the success of the application to the government, farmers, industrial producers, and consumers.

**Author Contributions:** Conceptualization, A.B., A.S., L.B. and H.C.-S.; methodology, A.B., A.S., H.C.-S., L.B. and F.N.A.; software, A.B. and A.C.B.; validation, A.S., H.C.-S. and L.B.; formal analysis, A.B., A.C.B. and L.B.; investigation, A.B., A.S., H.C.-S., F.N.A., L.B. and A.C.B.; resources, L.B. and F.N.A.; data curation, A.B. and A.C.B.; writing—original draft preparation, A.B., A.C.B. and L.B.; writing—review and editing, A.C.B. and L.B.; visualization, A.B., L.B., A.S., H.C.-S. and F.N.A.; supervision, A.S., H.C.-S. and L.B.; project administration, A.S. and L.B.; funding acquisition, F.N.A. All authors have read and agreed to the published version of the manuscript.

**Funding:** This research received no external funding.

**Institutional Review Board Statement:** Not applicable.

**Informed Consent Statement:** Not applicable.

**Data Availability Statement:** Not applicable.

**Conflicts of Interest:** The authors declare no conflict of interest.

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
