# Peer review of "Recent Advances in Encapsulation Techniques of Plant Growth-Promoting Microorganisms and Their Prospects in the Sustainable Agriculture"

_applsci, doi:10.3390/app12189020_

Round 1

Author Response

First of all,

The authors would like to thank editor and reviewers for their comments and efforts to improve our manuscript. We prepared a letter to reviewers comments and added our responses to them as below :

Reviewer 1

Open Review

English language and style

( ) Extensive editing of English language and style required
( ) Moderate English changes required
(x) English language and style are fine/minor spell check required
( ) I don't feel qualified to judge about the English language and style

Is the work a significant contribution to the field?

Is the work well organized and comprehensively described?

Is the work scientifically sound and not misleading?

Are there appropriate and adequate references to related and previous work?

Is the English used correct and readable?

  1. On Page 4, line 141 and 143: are the 10) self-assembly methods and 16) self-assembly the same or just repetition?

Corresponding Author : self-assembly and self-assembly methods are two different techniques according to Hudson and Margaritis (2014),  self-assembly is prepared by aggregates formation containing serum albumin as inner core and poly ethylene glycol as hydrophilic part of the micelles, self-assembly methods are multiple and include diafiltration method, self-assembly method assisted by sonication and solvent diffusion method, are formed by grafting copolymers onto dextran molecules. See reference « Hudson, D., & Margaritis, A. (2014). Biopolymer nanoparticle production for controlled release of biopharmaceuticals. Critical Reviews in Biotechnology34(2), 161-179. »

  1. On Page 10, line 381: Could the authors explain more about morphological parameters and how important they are?

Corresponding Author : Modifications has been added to the text. The morphological parameters reflects the physical appearance of the plant and it it concerns roots, shoots and leaves growth and morphology and stem architecture. Researchers focus on measuring morphological parameters after inoculation because they are considered plant growth factors and indicate yield improvement

  1. Table 1 is very clear and informative

Corresponding Author : Thank you for the comment

  1. It would be very helpful if the authors could also provide Information about the composition of each major component in the bioformulation of encapsulation in general. For example, what is the percentage of each additive? Is there any negative effect if certain additive is not enough or adding too much?

Corresponding Author: Sodium alginate, chitosan, chitin, amidon acide humique chemical composition and properties are mentionned in the text, and additives pourcentage has been added.

Additives are generally added and combined according to previous literature, some additives are added at low concentration (0.1-0.5%) like nanoparticles while others are added at high concentrations. Certain additives added at high concentrations can alter the viscosity thus affecting the morphologies of the beads and the release properties. Various research works has been devoted to studying the effect of concentration on the development of bioformulations. infrared spectroscopy and X-ray diffraction (XRD)  analyzes are carried out in this context. See references « Rohman, S., Kaewtatip, K., Kantachote, D., & Tantirungkij, M. (2021). Encapsulation of Rhodopseudomonas palustris KTSSR54 using beads from alginate/starch blends. Journal of Applied Polymer Science138(12), 50084. »

« Wu, Z., Guo, L., Qin, S., & Li, C. (2012). Encapsulation of R. planticola Rs-2 from alginate-starch-bentonite and its controlled release and swelling behavior under simulated soil conditions. Journal of Industrial Microbiology and Biotechnology39(2), 317-327. »

What is the size distribution of those encapsulation techniques? Does the size uniformity of encapsulation affect their effectiveness or viability? Are there any size control methods currently being applied in academia or industry?

Corresponding Author : The size of the particles generated essentially depends on the techniques used, spray drying technique generates fine powders (beads of a few micrometers in diameter), while ionic gelation and emulsification give capsules of different sizes depending on the equipment used, the encapsulator equipement gives beads of several micrometers in diameter while the needles give capsules of several millimeters (already explained in the manuscript). Letterature review reported that microparticles are advantageous and favored over macroparticles in terms of viability and field efficiency (discussed in the manuscript). According to our observations, ionic gelation is the most commonly used technique among researchers in academia and no size control method has been reported, while industries generally employ spray drying which generates particles of controlled size.

Reviewer 2 Report

The manuscript entitles “Recent advances in encapsulation techniques of plant growth promoting microorganisms and their prospects in the sustainable agriculture” has been written well and have some comments below:

In abstract please rewrite and concise this sentence of Line no 21.

In intro part, Line no 123. please give full stop after the sentence.

Line no 147. Full stop should be come after Figure 2.

Line no 160. Give space in between CaCl2[20].

Which encapsulation technique is best suitable and cost effective for the researchers, farmers and students used in agriculture system?

Font size should be increase for the figure 4 as content are not visible clearly.

Please put one table for Nanotechnological approach used for this bioencapsulation.

Kindly put outcome of this review in conclusion part specially which encapsulation technique, additive etc are very much useful and why?

References should be according to the journals guideline. And follow same format for all the references. For example Line no 602, ref no 15, The font size should be in small letters as I do in bold like, . Superior Polymeric Formulations and Emerging Innovative Products of Bacterial Inoculants for Sustainable Agriculture and the Environment”””

English language must be improved in throughout the manuscript.

Author Response

Reviewer 2

Open Review

English language and style

( ) Extensive editing of English language and style required
(x) Moderate English changes required
( ) English language and style are fine/minor spell check required
( ) I don't feel qualified to judge about the English language and style

Is the work a significant contribution to the field?

Is the work well organized and comprehensively described?

Is the work scientifically sound and not misleading?

Are there appropriate and adequate references to related and previous work?

Is the English used correct and readable?

Comments and Suggestions for Authors

The manuscript entitles “Recent advances in encapsulation techniques of plant growth promoting microorganisms and their prospects in the sustainable agriculture” has been written well and have some comments below:

In abstract please rewrite and concise this sentence of Line no 21.

Corresponding Author : It is done

In intro part, Line no 123. please give full stop after the sentence.

Corresponding Author : It is done

Line no 147. Full stop should be come after Figure 2.

Corresponding Author : It is done

Line no 160. Give space in between CaCl2[20].

Corresponding Author : It is done

Which encapsulation technique is best suitable and cost effective for the researchers, farmers and students used in agriculture system?

Corresponding Author : According to our findings, the techniques adopted for the encapsulation of most PGPMs in academia among researchers and students depends on the means and equipment available, most african/asian research teams have used the technique of cross linking (ionic gelation) which only requires a cations solution and a syringe (See in table 1 cross linking is the most reported). In developed countries researchers generally use spray drying as they have the necessary equipment. ionic gelation by syringe although it is cost and time effective, it is not suitable for large production intended for large-scale application, for this, the installation of encapsulators and laminar flow hood is required

Font size should be increase for the figure 4 as content are not visible clearly.

Corresponding Author :

Please put one table for Nanotechnological approach used for this bioencapsulation.

Corresponding Author : It is done

Kindly put outcome of this review in conclusion part specially which encapsulation technique, additive etc are very much useful and why?

Corresponding Author : It is done

References should be according to the journals guideline. And follow same format for all the references. For example Line no 602, ref no 15, The font size should be in small letters as I do in bold like, . Superior Polymeric Formulations and Emerging Innovative Products of Bacterial Inoculants for Sustainable Agriculture and the Environment”””

Corresponding Author: It is done.

English language must be improved in throughout the manuscript.

Corresponding Author: It is done.

Reviewer 3 Report

The review summarizes the knowledge of the scientific literature on encapsulation techniques of plant growth promoting microorganisms This review is well detailed and structured. However, these are some minor remarks

-          In the abstract, line 26, a space is needed.

-          In line 219, what does [2011] indicate?

-          In figure 4, the font size is smaller compared to other figures.

-          Are there some studies in the literature on the toxicity and impact of nanoparticles on the environment and on humans?

-          Kindly check the doi missing in some references 14,16,17,20, 86. 

Author Response

Reviewer 3

Open Review

English language and style

( ) Extensive editing of English language and style required
( ) Moderate English changes required
(x) English language and style are fine/minor spell check required
( ) I don't feel qualified to judge about the English language and style

Is the work a significant contribution to the field?

Is the work well organized and comprehensively described?

Is the work scientifically sound and not misleading?

Are there appropriate and adequate references to related and previous work?

Is the English used correct and readable?

Comments and Suggestions for Authors

The review summarizes the knowledge of the scientific literature on encapsulation techniques of plant growth promoting microorganisms This review is well detailed and structured. However, these are some minor remarks

-          In the abstract, line 26, a space is needed.

Corresponding Author : It is done

-          In line 219, what does [2011] indicate?

Corresponding Author : The reference has been modified.

-          In figure 4, the font size is smaller compared to other figures.

Corresponding Author :

-          Are there some studies in the literature on the toxicity and impact of nanoparticles on the environment and on humans?

 Corresponding Author : There are numerous studies in the literature on the toxicity and impact of nanoparticles on the environment and on humans. In reality, the harmful effect of nanomaterials and their risk of toxicity depend on their properties of reactivity, stability, mobility, chemical composition, their surface properties, solubility, their aggregation behavior, their biokinetics and their biopersistence. The dosage and concentration of nanoparticles and their chemical properties have been designed and synthesized in such a way as not to have a negative effect on the ecosystem. The question of safety of nanoparticles has already been addressed before their involvement in human medicine, particularly in bioimaging and anticancer therapies. However, the effect of their persistence in the environment and in food and feed is unknown, we have mentioned in outlook that this needs to be urgently investigated. See references

« Singh, R. P., Handa, R., & Manchanda, G. (2021). Nanoparticles in sustainable agriculture: An emerging opportunity. Journal of Controlled Release329, 1234-1248. »

« Hazarika, A., Yadav, M., Yadav, D. K., & Yadav, H. S. (2022). An overview of the role of nanoparticles in sustainable agriculture. Biocatalysis and Agricultural Biotechnology, 102399. »

« Sajid, M., Ilyas, M., Basheer, C., Tariq, M., Daud, M., Baig, N., & Shehzad, F. (2015). Impact of nanoparticles on human and environment: review of toxicity factors, exposures, control strategies, and future prospects. Environmental Science and Pollution Research22(6), 4122-4143. »

Kindly check the doi missing in some references 14,16,17,20, 86. 

Corresponding Author : References 14, 17 and 86 have no doi.